# Defense Mechanisms Induced by Celery Seed Essential Oil against Powdery Mildew Incited by *Podosphaera fusca* in Cucumber

**DOI:** 10.3390/jof10010017

**Published:** 2023-12-27

**Authors:** Hajar Soleimani, Reza Mostowfizadeh-Ghalamfarsa, Mustafa Ghanadian, Akbar Karami, Santa Olga Cacciola

**Affiliations:** 1Department of Plant Protection, School of Agriculture, Shiraz University, Shiraz 7144113131, Iran; hajarsoleimani97@gmail.com; 2Department of Pharmacognosy, Isfahan Pharmaceutical Sciences Research Center, Isfahan University of Medical Sciences, Isfahan 8174673461, Iran; ghannadian@gmail.com; 3Department of Horticultural Science, School of Agriculture, Shiraz University, Shiraz 7144113131, Iran; akbarkarami@shirazu.ac.ir; 4Department of Agriculture, Food and Environment (Di3A), University of Catania, 95123 Catania, Italy

**Keywords:** *Apium graveolens*, *Cucumis sativus*, d-limonene, phenols, flavonoids, chlorophyll, β-1,3-glucanase, chitinase, phenylalanine ammonia-lyase, gene expression

## Abstract

This study aimed to evaluate the effectiveness of essential oil extracted from celery (*Apium graveolens*) seeds (CSEO) for the control of powdery mildew of cucumber (*Cucumis sativus*) incited by *Podosphaera fusca* and to investigate the metabolic and genetic defense mechanisms triggered by the treatment with this essential oil in cucumber seedlings. The main compounds in the CSEO as determined by gas chromatography–mass spectrometry (GC-MS) analysis were d-limonene, 3-butyl phthalide, β-selinene, and mandelic acid. The treatment with CSEO led to an increase in the content of both chlorophyll and phenolic/flavonoid compounds in cucumber leaves. In greenhouse tests, the application of CSEO reduced by 60% the disease severity on leaves of cucumber plants and stimulated the activity of defense-related enzymes such as β-1,3-glucanase, chitinase, phenylalanine ammonia-lyase, peroxidase, and polyphenol oxidase. Moreover, treatment with CSEO induced overexpression of β-1,3-glucanase, chitinase, and phenylalanine ammonia-lyase genes. A highly significant correlation was found between the β-1,3-glucanase, chitinase, and phenylalanine ammonia-lyase enzymatic activities and the relative expression of the corresponding encoding genes in both inoculated and non-inoculated cucumber seedlings treated with the essential oil. Overall, this study showed that CSEO is a promising eco-friendly candidate fungicide that can be exploited to control cucumber powdery mildew.

## 1. Introduction

Powdery mildew is one of the most important diseases of cucurbits, including cucumber (*Cucumis sativus* L.) [1]. The most common causal agent of cucumber powdery mildew is *Podosphaera fusca* (Fr.) U. Braun & Shishkoff [syn. *Sphaerotheca fuliginea* (Schlecht Ex Fr.) Pollacci], in the family *Erysiphaceae* [2,3]. This biotrophic pathogen has a worldwide distribution and attacks cucumber leaves, stems, and fruits in both greenhouse and field cultivations [3]. Powdery mildew infections interfere with photosynthesis and respiration, leading to yield reduction, incomplete ripening, and poor taste of the product. Yield losses caused by powdery mildew of cucumber have been estimated at around 25% in India and up to 60% in Pakistan [4,5]. Presently, synthetic chemical fungicides are mainly used for the control of this disease. Although synthetic fungicides are effective against powdery mildew, their use on a large scale may have detrimental effects on human health and environment. Moreover, there is the risk of development of fungicide resistance by the pathogen. Consequently, in the last few years, research has focused on control methods alternative to synthetic fungicides [6].

Bioactive natural products of various origin and nature have been taken into consideration as an eco-friendly alternative to synthetic chemicals for crop protection [7,8,9,10,11]. Many plant-derived compounds, including essential oils, have been proven to possess antifungal activity. Plant essential oils are volatile organic compounds (VOCs) of terpenic and alcoholic nature. Most of them have a wide range of biological properties, such as antioxidant, antifungal, and antibacterial activities, that have been exploited in the food, cosmetic, and pharmaceutical industries [12]. Their efficacy in controlling fungal plant pathogens has been proven in various studies [13,14,15,16]. Also, it has been demonstrated that essential oils may induce plant defense-messenger molecules against pathogen attacks [17]. Some plant essential oil components, including indole acetic acid, flavonoid, and phenolic compounds, prevent plant diseases by activating plant defense responses such as the priming or increase in the production of enzymes and proteins related to pathogenicity. The accumulation of pathogenicity-related (PR) proteins in plants, such as β-1,3-glucanase (PR2), chitinase (PR3), and PR1, are regarded as markers of systemic induced resistance [18,19].

Celery (*Apium graveolens* L.), a plant in the family *Apiaceae*, has been cultivated as a vegetable since ancient times and has a worldwide distribution. It is a rich source of bioactive organic substances, such as alkaloids, carbohydrates, flavonoids, glycosides, and steroids [20]. The antibacterial and antifungal activity of celery leaf extract on plant pathogens, such as the bacteria *Pseudomonas aeruginosa* (Schroeter) Migula and *Staphylococcus epidermidis* (Winslow and Winslows) Evans, as well as the fungi *Heterobasidion annosum* (Fr.) Bref. and *Rhizoctonia* sp., has been previously recognized [21]. Aqueous, methanolic, and acetonic extracts of celery leaves were effective in controlling cucumber powdery mildew [22].

The objectives of the present study were to (*i*) test the effectiveness of celery seed essential oil (CSEO) in controlling cucumber powdery mildew; (*ii*) analyze the CSEO to identify its bioactive components; and (*iii*) study the effects of CSEO on the metabolism of cucumber plants and, in particular, its ability to induce defense mechanisms against powdery mildew.

## 2. Materials and Methods

### 2.1. Celery Seeds and Essential Oil Extraction

Celery seeds were purchased from Pakan Bazr (Isfahan, Iran) and were identified as *Apium graveolens* seeds by the Department of Pharmacognosy, Isfahan University of Medical Sciences, Iran. A seed voucher specimen (Sam-1704) was deposited at the Department of Pharmacognosy, Isfahan University.

Seeds (120 g) were powdered and transferred into a flask containing 1.2 L of distilled water (dw). The essential oil was extracted according to the hydrodistillation method described by Khamesipour et al. [23] using a Clevenger apparatus, for seven consecutive cycles, each of 3 h. The essential oil extracted from seeds (6.2 mL) was dehydrated with anhydrous sodium sulfate and stored in dark-colored vials at 4–6 °C.

### 2.2. Chemical Analysis of Essential Oil from Celery Seeds (CSEO)

The chemical analysis of the essential oil was performed by gas chromatography-mass spectrometry (GC-MS) using an Agilent 7890A GC coupled with an Agilent 5975C mass detector (Agilent Technologies, Santa Clara, CA, USA). One μL of CSEO was injected into the GC-MS and the metabolites were identified by calculating the retention index (RI) and mass spectrum. The RIs were compared with those reported in the literature in the WebBook NIST and Wiley275.L libraries. The analysis was performed by a GC-MS HP-5 column (30 m × 0.25 mm; film thickness 0.25 μm), with helium as a carrier, a 2 mL min^−1^ flow rate and 70 electron volts ionization energy. The initial oven temperature was 110 °C, held for 2 min and subsequently raised up to 270 °C by 3 °C min^−1^, for a total time of 57 min [23].

### 2.3. Effectiveness of CSEO in Controlling Cucumber Powdery Mildew

Cucumber seeds (*Cucumis sativus* cv. Super Dominus) were sowed in 14 cm plastic pots containing a mixture of soil, sand, and peat moss (1:1:1). The pots were kept in the greenhouse at 22 to 28 °C, with a relative humidity of 70 to 80%, and with a 14 h daylight cycle. When the second true leaves of the plants were developed, pots with uniformly vigorous seedlings were selected for the experiment. *Podosphaera fusca* HS101 strain was used to inoculate the seedlings. This powdery mildew strain was sourced in 2019 from cucumber plants with natural infections in Isfahan, Iran, and maintained with successive transfers on cucumber plants under greenhouse. It had been identified on the basis of morphological features according to Braun et al. [24]. Freshly infected leaves carrying conidia were picked from cucumber plants and immersed in dw containing 1 mL L^−1^ Tween 20. The concentration of conidia in the suspension was adjusted to 4 × 10^4^ mL^−1^, using a Bright-Line hemocytometer (HBG, Gießen, Germany) [25]. Seedlings were inoculated by spraying the conidial suspension on the leaves. The inoculated seedlings were kept under a dark plastic cover and humidity above 90% for 24 h, and then returned to greenhouse conditions. Disease symptoms appeared 8 to 10 days after inoculation.

To evaluate the efficacy in controlling powdery mildew, CSEO was dispersed at concentrations of 100, 200, and 400 μg mL^−1^ in dw containing 1 mL L^−1^ of Tween 20 and the emulsion was sprayed on cucumber seedlings either one day before the inoculation or one day after the inoculation. Control plants were sprayed only with dw containing 1 mL L^−1^ of Tween 20. The experiment was arranged in a complete randomized block with four replicates. Single replicates consisted of four pots, each with one seedling. The experiment was repeated once. However, as results of the two experiments were very similar, only results of one experiment are reported.

The disease severity was assessed 10 days post inoculation (dpi). Symptoms were rated using the scale of Masyita et al. [12], based on the proportion of leaf area covered by the typical powdery mildew colonies. The scale includes the following steps: 0 = asymptomatic, 1 = 1–5% of the leaf area infected, 2 = 6–25% of the leaf area infected, 3 = 26–50% of the leaf area infected, 4 = more than 50% of the leaf area infected, and 5 = dead leaf. The disease severity was calculated based on the following formula:Disease severity (%) = [∑ (n × v)/5N] ×100

In this formula, n = the number of infected leaves in a plant, v = the disease severity of each leaf according to the scale from 0 to 5, and N = total number of leaves in a plant.

### 2.4. Effects of Treatment with CSEO on the Physiology of Cucumber Seedlings and Induction of Defense Mechanisms against Powdery Mildew

#### 2.4.1. Experimental Design

The effect of the treatment with CSEO on the plant physiology and the induction of defense mechanisms against *P. fusca* infection was studied in a separate experiment on artificially inoculated cucumber seedlings. Seedlings at the four-leaf stage were divided into four sets. The seedlings of the first set were inoculated with *P. fusca* as described previously and, soon after the onset of symptoms (10 days after inoculation), were sprayed with an emulsion of CSEO (400 μg mL^−1^) in dw containing 1 mL L^−1^ of Tween 20. Simultaneously, non-inoculated seedlings of a second set were sprayed with the CSEO emulsion (400 μg mL^−1^), while seedlings of the third and fourth sets, comprising inoculated and non-inoculated seedlings, respectively, were sprayed with dw containing Tween 20. At various time intervals after the treatment with CSEO (zero, i.e., within one hour after the treatment, and one, two, four, and eight days after the treatment), leaf samples were taken from seedlings of all sets and processed. The experiment was arranged in a complete randomized block with four replicates, each consisting of four pots with one seedling. The experiment was performed twice. However, as results of the two experiments were very similar, only the results of one experiment are reported.

#### 2.4.2. Determination of Chlorophyll, Phenolic, and Flavonoid Compounds in Cucumber Leaves

The chlorophyll content of cucumber leaves was quantified using the method of Aron [26]. The amount of phenolic compounds (mg of gallic acid per g of leaf fresh weight) was measured by the Folin–Ciocalteu reagent (Sigma-Aldrich, St. Louis, MO, USA) and methanolic extract following the method of Maizura et al. [27]. Flavonoid compounds (quercetin per g of leaf fresh weight) were determined by the method described by Chang et al. [28] using methanolic extract.

#### 2.4.3. Effect of CSEO on Activity of Defense Enzymes in Cucumber Leaves

The enzyme activities of β-1,3-glucanase, chitinase, phenylalanine ammonia-lyase, peroxidase, and polyphenol oxidase in cucumber leaves were analyzed using the methods described by Yedidia et al. [29], Khan & Umar [30], Goldson et al. [31], Monsur et al. [32], and Kar & Mishra [33], respectively.

#### 2.4.4. Gene Expression of β-1,3-Glucanase, Chitinase, and Phenylalanine Ammonia-Lyase in Cucumber Leaves

##### RNA Extraction

Total RNA from the leaves was extracted with a total RNA isolation kit (Denazist Asia, Mashhad, Iran) based on the manufacturer’s instructions. The quantity and quality of the extracted RNA were evaluated on 1% agarose gel electrophoresis and by a NanoDrop spectrophotometer (BioTek Instruments Inc., Winooski, VT, USA). About 0.4 μg of total extracted RNA was used for cDNA synthesis.

##### cDNA Synthesis

cDNA synthesis was performed using a cDNA synthesis kit (Addbio, Daejeon, Republic of Korea) according to the manufacturer’s instructions. The sequence of genes was obtained from the GenBank database [34], and specific primers were designed using the Primer3 software 0.4.0 [35]. The PCR product was visualized on Qubite 4 Fluorometer (Invitrogen, Singapore).

##### Quantitative Real-Time PCR Assay

A quantitative real-time polymerase chain reaction (RT-qPCR) was performed using SYBR green Master Mix (Isfahan University of Medical Sciences, Core Research Facilities, Isfahan, Iran). The actin gene was used as an internal control gene. The reaction volume was 15 μL, containing 1.5 μL of cDNA, 7.5 μL of SYBR green Master Mix, 0.3 μL of each of the forward and reverse primers, and 5.4 μL of RNase-free water. The thermal cycling program for PCR amplification for all genes was 95 °C, 10 min; 40 cycles of 95 °C, 10 s; 61 °C, 10 s; and 72 °C, 15 s. The level of gene expression was measured using the method described by Livak & Schmittgen [36]. The experiment was performed for each treatment at each time span with three replications.

### 2.5. Statistical Analysis

The two-way ANOVA (analysis of variance) was used for statistical analysis of the data generated in the experiments. For experiments involving two variables, where one was not statistically significant according to two-way ANOVA, the insignificant variable was eliminated. The data analysis was then repeated using one-way ANOVA. The data normality was verified with the Shapiro–Wilk test and percentages were transformed into Arcsine. Means were compared using the Least Significant Difference (LSD) or Tukey’s honestly significant difference (HSD). For statistical analysis, the data concerning gene expression were transformed into natural logarithms and the significance of the differences was tested using the Tukey’s multiple comparison test.

Pearson’s correlation was used to determine the relationship between the enzyme activities and the corresponding gene expressions. The heatmap representing the gene expression in leaves of infected and CSEO-treated cucumber seedlings was drawn using the relative expression (fold change) of each gene compared to untreated healthy seedlings at the diverse sampling intervals. Data were analyzed by the R software, version 4.2.1.

## 3. Results

### 3.1. Chemical Composition of CSEO

The results of the GC-MS analysis of the CSEO are shown in Table 1 and Appendix A. Overall, 11 compounds were identified. The main CSEO components were d-limonene, (42.74%), 3-butyl phthalide (14.42%), β-selinene (11.91%), and mandelic acid (10.92%). Other compounds accounted for the remaining 10.01%.

### 3.2. Efficacy of CSEO in Controlling Cucumber Powdery Mildew

No statistically significant effect of scheduled treatments (before or after artificial inoculation of cucumber seedlings with the pathogen) on disease severity was observed (Appendix A). In addition, there was no significant effect of the interaction between scheduled treatments and CSEO concentrations on the disease severity. By contrast, there was a significant (*p* ≤ 0.001) effect of CSEO concentrations on disease severity. All CSEO concentrations significantly reduced, although to different extent, the cucumber powdery mildew severity compared to the control. The concentration of 400 μg mL^−1^ was the most effective. Figure 1 shows that the powdery mildew severity in seedlings treated with CSEO at 400 μg mL^−1^ one day after the inoculation compared with the untreated control was 41.25%. Conversely, it was 75% and around 70% in seedlings treated with 100 and 200 μg mL^−1^ of CSEO, respectively, with no significant difference between these two concentrations (Figure 1).

### 3.3. Effects of the Treatment with CSEO on the Physiology of Cucumber Seedlings and Its Ability to Induce Defense Mechanisms against Powdery Mildew

#### 3.3.1. Chlorophyll Content in Cucumber Leaves

The two-way ANOVA statistical analysis showed that the effect of different sampling intervals, as well as of the interaction between sampling intervals and type of treatment received by the four cucumber seedlings sets on chlorophyll concentration, was highly significant (Appendix A). At all sampling intervals, the chlorophyll content in leaves of the two sets inoculated with powdery mildew was lower than the chlorophyll content in leaves of the non-inoculated sets, as expected (Figure 2A). Moreover, throughout the experiment, the chlorophyll concentration increased progressively in the leaves of all seedlings but those inoculated with powdery mildew and untreated with CSEO.

The chlorophyll concentration in leaves of inoculated cucumber seedlings treated with CSEO was significantly (*p* ≤ 0.001) higher than in untreated inoculated seedlings at all sampling intervals after the treatment (Figure 2A). The most relevant difference was observed 8 d after the application of CSEO (Figure 2A). In leaves of inoculated seedlings treated with CSEO, the chlorophyll concentration at 0, 1, 2, 4, and 8 d after the treatment was 12.52, 85.26, 106.88, 146.84, and 875.54% higher, respectively, than in leaves of inoculated but untreated seedlings.

In leaves of non-inoculated seedlings treated with CSEO, the chlorophyll concentration at 0, 1, and 2 d after the application of essential oil was significantly (*p* ≤ 0.001) higher than in leaves of non-inoculated and untreated seedlings (Figure 2A). The difference was of 24.57, 15.77, and 4.34%, respectively. Conversely, at 4 and 8 d after the treatment the chlorophyll concentration in leaves of non-inoculated seedlings treated with CSEO was significantly lower (8.7 and 15.98%, respectively) than in leaves of untreated seedlings, suggesting a slight toxic effect of CSEO.

#### 3.3.2. Content of Phenolic Compounds in Cucumber Leaves

The two-way ANOVA statistical analysis revealed that the sampling times as well as the interaction between sampling times and the treatment received by cucumber seedlings had significant effects (*p* ≤ 0.001) on the amount of phenolic compounds in cucumber leaves (Appendix A). Overall, the total phenolic content was significantly (*p* ≤ 0.001) higher in leaves of seedlings inoculated with powdery mildew compared with the leaves of non-inoculated seedlings, irrespective of the type of treatment (dw versus CSEO) (Figure 2B and Appendix A). In leaves of inoculated seedlings treated with CSEO, the total phenolic content peaked soon after the treatment (sampling time 0) and initially was higher than in leaves of untreated inoculated seedlings. However, it decreased progressively but noticeably throughout the experiment. By contrast, in untreated seedlings, it increased progressively, and, at the two last sampling times (4 and 8 d after the treatment), there was no significant difference between treated and untreated seedlings (Figure 2B). A similar trend was observed in leaves of non-inoculated seedlings treated with CSEO. In particular, in leaves of seedlings inoculated with powdery mildew and treated with CSEO, at 0, 1, and 2 d sampling times, the content of phenolic compounds increased by 71.12, 26.14, and 14.40%, respectively, compared to the inoculated but untreated seedlings (Figure 2B).

#### 3.3.3. Content of Flavonoid Compounds in Cucumber Leaves

Statistical analysis revealed that the sampling times, as well as the interaction between sampling times and the treatment received by cucumber seedlings, had significant effects (*p* ≤ 0.001) on the amount of flavonoid compounds in cucumber leaves (Appendix A). In leaves of inoculated cucumber seedlings treated with CSEO, the flavonoid content was significantly (*p* ≤ 0.001) higher than in leaves of inoculated but untreated seedlings at all sampling times but one (4 d after the treatment) (Figure 2C and Appendix A). Similarly, as for the two sets of non-inoculated seedlings, the flavonoid content in the leaves of seedlings treated with CSEO was significantly (*p* ≤ 0.001) higher than in leaves of untreated seedlings at all sampling times but one (0 d after the treatment) (Figure 2C). In particular, at 1, 2, 4, and 8 d after the application of essential oil, the content of flavonoid compounds in leaves of these seedlings was 38.83, 40.89, 33.34, and 31.04% higher than in untreated seedlings.

#### 3.3.4. Effect of CSEO on the Activity of Defense Enzymes against Cucumber Powdery Mildew

The two-way ANOVA statistical analysis revealed that the activities of β-1,3-glucanase, chitinase, phenylalanine ammonia-lyase, peroxidase, and polyphenol oxidase were significantly (*p* ≤ 0.001) affected by essential oil treatment in both inoculated and non-inoculated cucumber seedlings (Appendix A). It also indicated that the effects of sampling times and the interaction between sampling times and the treatments to which the four seedling sets were subjected (treated infected plants versus untreated infected plants and treated non-inoculated plants versus untreated non-inoculated plants) on the activities of these five enzymes were highly (*p* ≤ 0.001) significant (Appendix A).

##### β-1,3-Glucanase

The activity of this enzyme was significantly higher in CSEO-treated than in untreated seedlings (Figure 3A). Moreover, it was significantly higher in seedlings inoculated with powdery mildew than in non-inoculated seedlings (Figure 3A). In all four seedling sets, the activity of β-1,3-glucanase increased progressively throughout the experiment. The only exceptions were the non-inoculated seedlings treated with CSEO, which, after a progressive increase at 0, 1, 2, and 4 d after the treatment, showed a decline of this enzymatic activity at 8 d. The maximum peak of β-1,3-glucanase activity was reached 8 d after the treatment in leaves of seedlings inoculated with powdery mildew and treated with CSEO, with an activity 44.43% higher than in leaves of seedlings inoculated with powdery mildew but not treated with the essential oil.

##### Chitinase

Chitinase activity in leaves of cucumber seedlings showed a trend similar to that of β-1,3-glucanase, with few exceptions (Figure 3B). It was higher in seedlings inoculated with powdery mildew than in non-inoculated seedlings and increased after the treatment with CSEO. In inoculated CSEO-treated seedlings, the enzyme activity at 1, 2, and 4 d after the treatment increased by 26.54, 40.90, and 20.49% compared with the untreated seedlings. However, at 8 d after the treatment, no significant difference was observed between treated and untreated seedlings (Figure 3B). The maximum peak of chitinase activity was observed in leaves of seedlings inoculated with powdery mildew and treated with CSEO, 2 d after the treatment (Figure 3B). As for the two sets of non-inoculated seedlings, the chitinase activity was higher in CSEO-treated than in untreated seedlings at all sampling intervals.

##### Phenylalanine Ammonia-Lyase

Both the infection of powdery mildew and the treatment with CSEO stimulated the activity of phenylalanine ammonia-lyase in cucumber leaves (Figure 3C). In the two treated seedling sets, the activity of phenylalanine ammonia-lyase was higher in seedlings infected with powdery mildew at all sampling times (Figure 3C). In the two seedling sets treated with CSEO, the activity was significantly higher in powdery mildew-infected seedlings at the 0 and 1 d sampling intervals. At 2 d after the treatment, no statistically significant difference was observed between the two sets, while at 4 and 8 d, the activity was higher in non-inoculated seedlings (Figure 3C). At 0, 1, 2, 4, and 8 d after the treatment with CSEO, the phenylalanine ammonia-lyase activity in leaves of powdery mildew-infected seedlings treated with the essential oil was by 65.55, 56.06, 40.83, 39.59, and 21.10%, respectively, higher than in powdery mildew-infected but untreated seedlings and peaked 4 d after the treatment (Figure 3C).

##### Peroxidase

The activity of this enzyme was also stimulated by both the infection of powdery mildew and the treatment with CSEO (Figure 3D). In non-inoculated seedlings treated with CSEO, at 0, 1, 2, 4, and 8 d after the treatment, the enzyme activity was by 60.67, 96.11, 303.76, 459.78, and 596.69%, respectively, higher than in non-inoculated untreated seedlings (Figure 3D).

The maximum peak of peroxidase activity was detected in powdery mildew-infected seedlings treated with CSEO, at 8 d after the treatment (Figure 3D). However, powdery mildew-infected seedlings treated with CSEO showed a higher peroxidase activity than untreated seedlings only in an advanced stage of infection, i.e., 4 and 8 d after the treatment. At these two time intervals, the enzyme activity in leaves of treated seedlings was 21.78 and 33%, respectively, higher than in untreated seedlings.

##### Polyphenol Oxidase

Both the infection by powdery mildew and the treatment with CSEO stimulated the polyphenol oxidase activity (Figure 3E). In powdery mildew-infected seedlings treated with CSEO, polyphenol oxidase activity was significantly higher than in powdery mildew-infected but untreated seedlings at all sampling times with the only exception of the first and third one (0, and 2 d after the treatment). At these time intervals, the difference between treated and untreated seedlings was not significant (Figure 3E).

In non-inoculated seedlings, the treatment with CSEO significantly increased the polyphenol oxidase activity at all sampling intervals with a maximum at 8 d after the treatment (Figure 3E). However, the absolute highest peak of enzyme activity was detected at 8 d after the treatment in powdery mildew-infected seedlings treated with CSEO (Figure 3E).

#### 3.3.5. Gene Expression in Cucumber Leaves

##### β-1,3-Glucanase Gene

The treatment with CSEO induced an overexpression of the β-1,3-glucanase gene in leaves of cucumber seedlings (Figure 4(A1,A2)). At 2, 4, and 8 d after the treatment, the expression of this gene in powdery mildew-infected cucumber seedlings treated with CSEO was significantly (*p* ≤ 0.001) higher (1.6, 2.3, and 3.4-fold, respectively) than in untreated infected seedlings (Figure 4(A1)).

In leaves of CSEO-treated healthy seedlings, the expression of β-1,3-glucanase gene, 2 and 4 d after the treatment with essential oil, increased significantly (*p* ≤ 0.05) compared with the untreated healthy seedlings (Figure 4(A2)).

##### Chitinase Gene

Also, for this gene, the treatment with CSEO induced a significant overexpression in leaves of cucumber seedlings. At the 0, 1, 2, and 4 d sampling intervals, the treatment with essential oil increased the chitinase gene expression in powdery mildew-infected cucumber seedlings by 1.2, 1.7, 2.4, and 1.6-fold, respectively, compared to the untreated inoculated plants (Figure 4(B1)).

In CSEO-treated healthy seedlings, chitinase gene expression significantly increased at all sampling intervals but one compared to untreated seedlings. Only at 2 d after the treatment were the differences between treated and untreated seedlings not significant (Figure 4(B2)).

##### Phenylalanine Ammonia-Lyase Gene

Also, for the phenylalanine ammonia-lyase gene, the treatment with CSEO induced a significant overexpression in leaves of cucumber seedlings. At all sampling intervals after the application of CSEO (0, 1, 2, 4 and 8 d after the treatment), the expression of this gene in treated infected seedlings was 2.7, 2.6, 2.1, 2, and 1.4-fold higher, respectively, than in untreated infected seedlings (Figure 4(C1)).

Similarly, phenylalanine ammonia-lyase gene expression was significantly (*p* ≤ 0.001) increased at all sampling intervals in CSEO-treated healthy cucumber seedlings compared with the untreated healthy seedlings (Figure 4(C2)).

### 3.4. Correlation between the β-1,3-Glucanase, Chitinase, and Phenylalanine Ammonia-Lyase Enzyme Activities and the Expression of the Corresponding Genes

The expression of the genes investigated in this study was correlated with the activities of the corresponding enzymes, as expected (Figure 5). In seedlings inoculated with powdery mildew, a significant positive correlation was found between β-1,3-glucanase (r = 0.85, *p* ≤ 0.01), chitinase (r = 0.77, *p* ≤ 0.01) and phenylalanine ammonia-lyase (r = 0.75, *p* ≤ 0.05) activities and the expression of the corresponding encoding gene in cucumber leaves at all sampling intervals. Also, in non-inoculated seedlings, a significant positive correlation was found between β-1,3-glucanase (r = 0.76, *p* ≤ 0.01), chitinase (r = 0.94, *p* ≤ 0.001), and phenylalanine ammonia-lyase (r = 0.99, *p* ≤ 0.001) activities and the expression of the corresponding genes in cucumber leaves at all sampling intervals.

### 3.5. Heatmap of Gene Expression

In the heatmap analysis (Figure 6), the expression of β-1,3-glucanase, chitinase, and phenylalanine ammonia-lyase genes in all four seedling sets was divided into two clusters, including inoculated and non-inoculated sets, respectively, regardless of the treatment with the essential oil they received. The gene expressions in all sampling intervals were also divided into two major clusters. The expression of chitinase and phenylalanine ammonia-lyase genes each fell in a main cluster comprising several sub-clusters. In all CSEO-treated and -untreated seedlings inoculated with powdery mildew, the expression of the chitinase gene was higher than that of other genes, while in the CSEO-treated non-inoculated plants, the expression of the phenylalanine ammonia-lyase gene was higher than that of other genes.

## 4. Discussion

This study demonstrated the efficacy of CSEO as a natural, eco-friendly, and safe compound for the control of cucumber powdery mildew caused by *P. fusca* and showed that the treatment with this essential oil induced natural biochemical and molecular defense mechanisms in cucumber leaves. There are several studies dealing with the antifungal and antibacterial activity of essential oils against plant pathogens and plant diseases, including cucumber powdery mildew. For example, lemon, lemongrass, and thyme essential oils individually and in a mixture significantly reduced the incidence and severity of cucumber powdery mildew [1]. In the study of Mostafa et al. [1], the treatment with lemon essential oil reduced cucumber powdery mildew by around 74% but at a concentration more than five times higher than the concentration of CSEO used in the present study. Other essential oils have showed inhibitory activity against *Podosphaera* species, such as for instance *P. xanthii* (Castagne) U. Braun & Shishkoff [14,47]. However, none of these essential oils was from *Apiaceae* species.

The major components of CSEO as determined in this study are the monoterpene d-limonene, the sesquiterpene β-Selinene, as well as the two aromatic compounds 3-butylphthalide and mandelic acid. Consistently with these results, other studies reported the major CSEO components included β-myrcene, β-pinene, β-selinene, d-limonene, and 3-butylphthalide [48,49,50]. However, the proportions of diverse components are not always consistent among diverse studies. Discrepancies might be due to the geographical origin, climatic and growing conditions, harvesting techniques, and chemotype and ontogeny of the plants used to extract the oil [51]. The variability in composition among different batches of the same essential oil can also explain inconsistencies in its antimicrobial activity [51].

The antifungal activity of essential oils seems to be largely related to the presence of terpenoid compounds [51]. The antifungal activity of d-limonene, the major terpenoid component of CSEO, was previously reported by Kalleli et al. [17] and Zorga et al. [52]. β-selinene and 3-butylphthalide were also reported to possess significant inhibitory activity against some ascomycetes such as *Aspergillus niger* Tieghem, and *Penicillium expansum* Link [52]. It is very probable that the inhibitory activity of CSEO against powdery mildews caused by *Podosphaera* species depends on the above-mentioned constituents, especially terpenoids.

The increase in chlorophyll concentration in leaves of cucumber seedlings infected with powdery mildew treated with CSEO can be explained, at least in part, by the reduction of disease severity as a consequence of the treatment. By contrast, the corresponding decrease in chlorophyll concentration in leaves of untreated seedlings infected with powdery mildew can be the result of the increase in disease severity during the experiment and the consequent loss of photosynthetic pigment. This trend of chlorophyll concentration in leaves of cucumber seedlings is consistent with the results of a previous study that tested the efficacy of other essential oils against cucumber powdery mildew [1]. Interestingly, however, in leaves of healthy cucumber seedlings treated with CSEO, initially (0, 1, and 2 d after the treatment) the chlorophyll concentration increased progressively and was significantly higher than in untreated seedlings, indicating that the treatment stimulated the chlorophyll synthesis. By contrast, at 4 and 8 d after the treatment, the chlorophyll concentration in leaves of treated cucumber seedlings was significantly lower than in leaves of untreated seedlings, suggesting a delayed phytotoxic effect of CSEO. Similar studies confirmed the negative effect of essential oils on the chlorophyll concentration of treated plants [53]. The interference of essential oil on the synthesis of chlorophyll deserves further investigation before its application as a fungicide in agriculture.

Overall, the results of this study showed that the treatment with CSEO induced a burst of defense mechanisms in cucumber seedlings. The content of phenolic compounds, expressed in terms of gallic acid concentration, increased significantly in leaves of both seedlings infected with powdery mildew and those treated with CSEO, compared with that of the untreated healthy controls. The increase in phenolic content in plant tissues in response to the essential oil treatment has been observed in other studies [17,54]. The increase in concentration of phenolic compounds is also a common plant defense response to fungal infections, and it is generally assumed that its effect is to limit the pathogen growth [54]. As phenols also have antioxidant properties [55], it can be hypothesized that the increase in these compounds in CSEO-treated healthy cucumber seedlings is a response to stress triggered by the treatment. The accumulation of phenolic compounds in infected plant tissues increases the mechanical robustness of cell walls, which results in a general resistance to biotic and abiotic stresses [17]. The progressive polymerization of phenolic compounds with other cell wall components may lead to a decrease in gallic acid concentration [56]. This metabolic process can explain the progressive reduction of gallic acid concentration observed in leaves of both CSEO-treated and infected cucumber seedlings.

After the treatment with CSEO, both the infected and healthy cucumber leaves showed a significant increase in concentration of flavonoid compounds compared with the leaves of untreated healthy plants. Many studies reported the increase in flavonoid compounds in response to essential oil treatment [17,57]. Conversely, the level of flavonoid compounds did not show any change in untreated seedlings infected with powdery mildew compared to healthy seedlings, except for at 1 d. According to these results, it seems that the infection by *P. fusca* did not provoke any increase in flavonoid concentration in cucumber leaves. This is inconsistent with the findings of other authors [58,59]. Flavonoid compounds play an important role in resistance of plants against pathogens and can prevent pathogen spread by stimulating tissue differentiation, leading in some plants to the generation of structures such as tyloses which restrict the spread of the pathogen within the vascular tissue [60]. Consistently with other studies [17,57], it can be hypothesized that the increase in flavonoid compounds in healthy cucumber seedlings treated with CSEO was a reaction to the oxidative stress caused by the essential oil.

The treatment with CSEO also enhanced the activity of enzymes typically involved in the plant defense reaction against biotic stresses [57,61,62,63,64]. Not surprisingly, a significant increase in β-1,3-glucanase, chitinase, phenylalanine ammonia-lyase, peroxidase, and polyphenol oxidase activities in leaves of untreated cucumber seedlings infected with powdery mildew compared with healthy untreated seedlings was observed. However, the CSEO-treated cucumber seedlings showed an even higher increase in these defense-related enzymatic activities compared with the corresponding untreated controls, confirming that the treatment with CSEO stimulated the plant defense mechanisms.

The synthesis of phenylpropanoid is a common response to biotic and abiotic stresses [65]. Increased activities of phenylalanine ammonia-lyase and other enzymes involved in the phenylpropanoid synthesis pathway, such as peroxidase and polyphenol oxidase [66], may be responsible for the accumulation of phenolic and flavonoid compounds in leaves of both CSEO-treated and powdery mildew-infected cucumber seedlings. Also, the increase in the peroxidase and polyphenol oxidase activities could have been related to the formation of oxygen-containing reactive species (ROS) causing the oxidation of phenolic compounds and their conversion into quinones [64]. The accumulation of phenolic and flavonoid compounds in plant tissues infected with pathogens is strictly related with ROS production mediated by peroxidase and polyphenol oxidase, and very probably its major function in infected plants is to inhibit the growth of the pathogen [60,61]. Other defense-related enzymes, such as β-1,3-glucanase and chitinase, exert a direct inhibitory activity against fungal plant pathogens or stimulate the production of phytoalexins and phenolic compounds in infected plants [64,67]. Results of this study are consistent with previous reports indicating the potential of essential oils to increase phenylalanine ammonia-lyase, peroxidase, polyphenol oxidase, β-1,3-glucanase, and chitinase activities in treated plants [57,62,64,67]. The stimulation of defense-related enzyme activities by essential oils in healthy plants is typically the result of the activation of antioxidant systems against oxidative stresses [53,65,68].

This study demonstrated that the increased activity of defense-related enzymes, such as β-1,3-glucanase, chitinase and phenylalanine ammonia-lyase in leaves of CSEO-treated cucumber seedlings was strictly correlated with the over-expression of the corresponding encoding genes. β-1,3-glucanase, chitinase, and phenylalanine ammonia-lyase are well-known defense-related genes whose expression in plants is stimulated by biotic and abiotic stresses [57,62,63]. An increase in the expression level of these genes after essential oil treatment was reported by other studies [57,62,67,69]. The heatmap depicting the expression level of β-1,3-glucanase, chitinase, and phenylalanine ammonia-lyase relative to untreated healthy cucumber seedlings confirmed that the highest expression level, especially for chitinase, occurred in leaves of infected cucumber seedlings treated with CSEO. This indicates the essential oil impacted the expression of genes involved in the defense mechanisms against cucumber powdery mildew.

Based on the findings of this study, CSEO as a natural compound can be an eco-friendly and safe alternative to synthetic chemical fungicides to control cucumber powdery mildew. Interestingly, this essential oil is effective at a low dose and stimulates general defense mechanisms in treated cucumber plants, suggesting a possible large-spectrum-activity against different fungal pathogens. However, other aspects should be addressed before practical application, including further investigation of its phytotoxicity and the search of a proper formulation.

## Figures and Tables

**Figure 1 jof-10-00017-f001:**
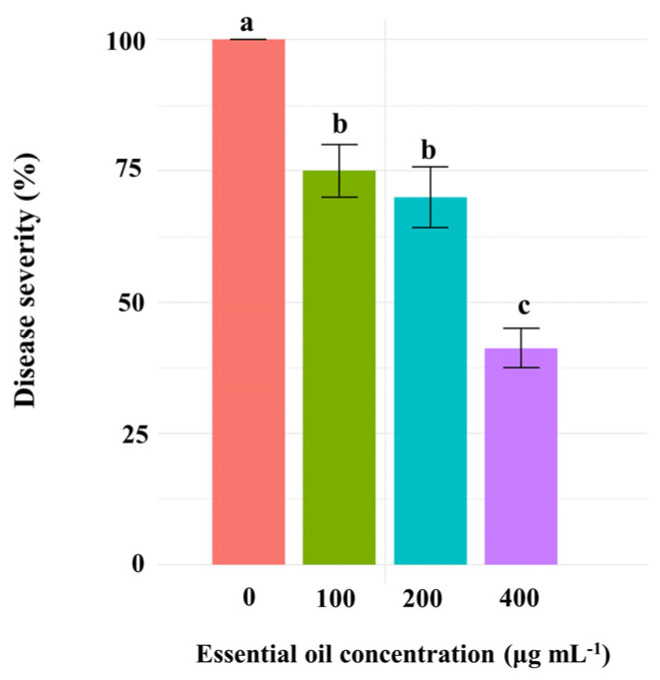
Severity of powdery mildew in cucumber seedlings sprayed with different concentrations of celery seed essential oil (100, 200, and 400 μg mL^−1^) one day after inoculation, compared to inoculated control seedlings treated with dw. Different letters indicate a significant difference for *p* ≤ 0.01 (LSD test).

**Figure 2 jof-10-00017-f002:**
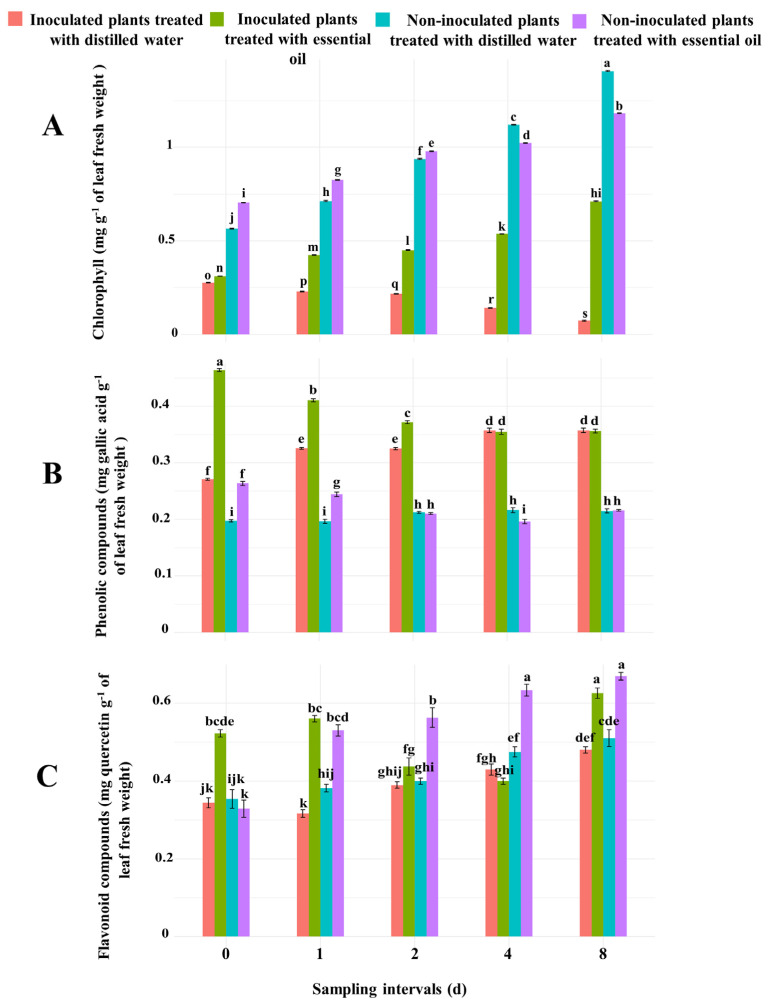
Effect of treatment with celery seed essential oil (400 μg mL^−1^) on the (**A**) chlorophyll content, (**B**) phenolic compounds, and (**C**) flavonoids of leaves of cucumber seedlings inoculated with powdery mildew compared to non-inoculated seedlings, at five different time intervals after the treatment. Different letters indicate significant differences for *p* ≤ 0.001 (Tukey’s test).

**Figure 3 jof-10-00017-f003:**
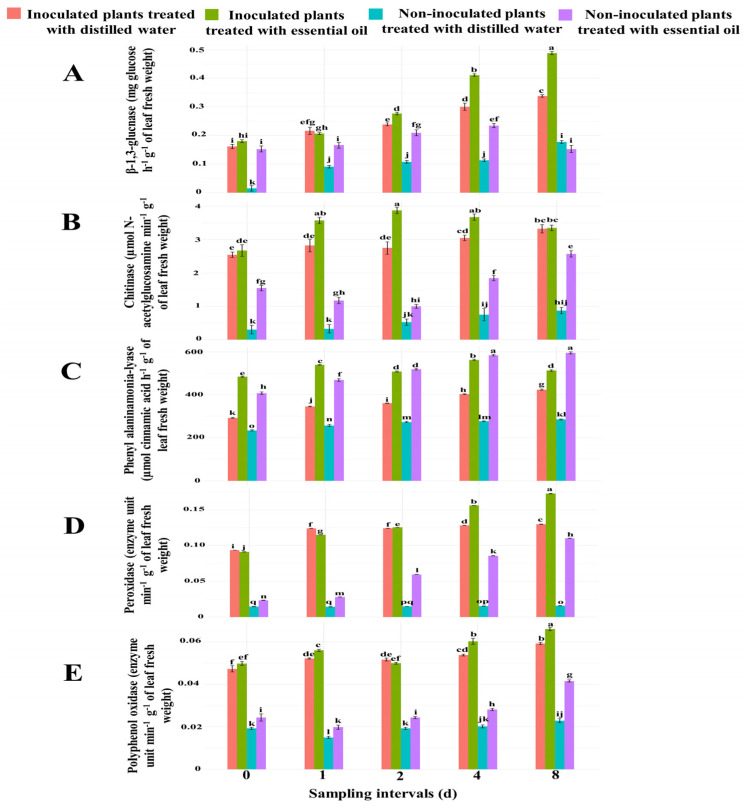
Effect of treatment with celery seed essential oil (400 μg mL^−1^) on (**A**) β-1,3-glucanase, (**B**) chitinase, (**C**) phenylalanine ammonia-lyase, (**D**) peroxidase, and (**E**) polyphenol oxidase activity in cucumber leaves either inoculated or non-inoculated with powdery mildew, compared to untreated seedlings, at five different time intervals after the treatment. Different letters indicate significant differences for *p* ≤ 0.001 (Tukey’s test).

**Figure 4 jof-10-00017-f004:**
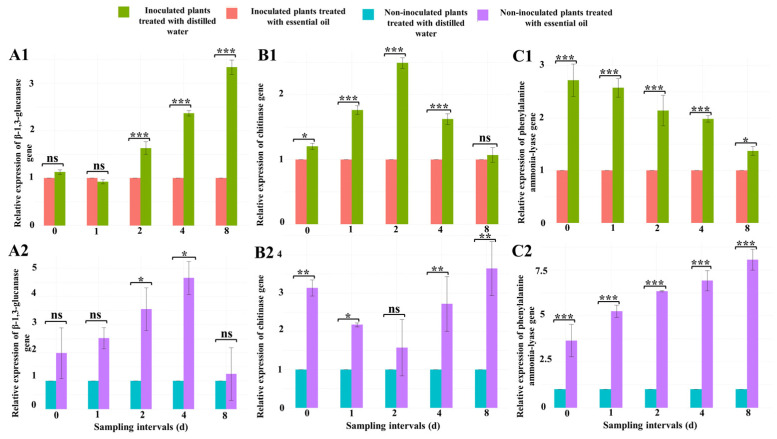
Effect of celery essential oil (400 μg mL^−1^) on the relative expression (fold change) of β-1,3-glucanase (**A1**,**A2**), chitinase (**B1**,**B2**), and phenylalanine ammonia-lyase (**C1**,**C2**) in the cucumber leaves inoculated and non-inoculated with powdery mildew at different sampling intervals after the treatment with essential oil. The treatment was carried out 10 d after inoculation soon after the onset of symptoms, and the zero point was within one hour after the treatment. Different symbols indicate significant differences between each set and the corresponding control set at the same time interval according to Tukey’s multiple comparison test. Significance symbols: 0 ***; 0.001 **; 0.01 * 0.05; ns non-significant.

**Figure 5 jof-10-00017-f005:**
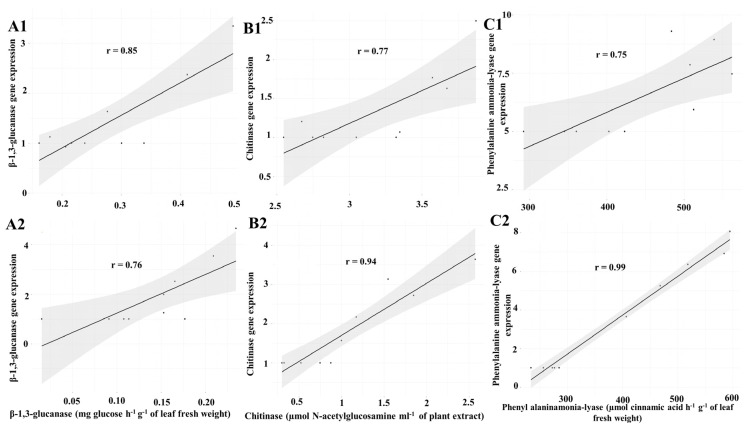
Correlation between the activities of β-1,3-glucanase (**A1**,**A2**), chitinase (**B1**,**B2**), and phenylalanine ammonia-lyase (**C1**,**C2**) and their corresponding gene expressions in cucumber leaves. The leaves were either inoculated (**A1**,**B1**,**C1**) or non-inoculated (**A2**,**B2**,**C2**) with cucumber powdery mildew. The correlation was determined at different sampling intervals after the treatment with celery seed essential oil at a concentration of 400 μg mL^−1^.

**Figure 6 jof-10-00017-f006:**
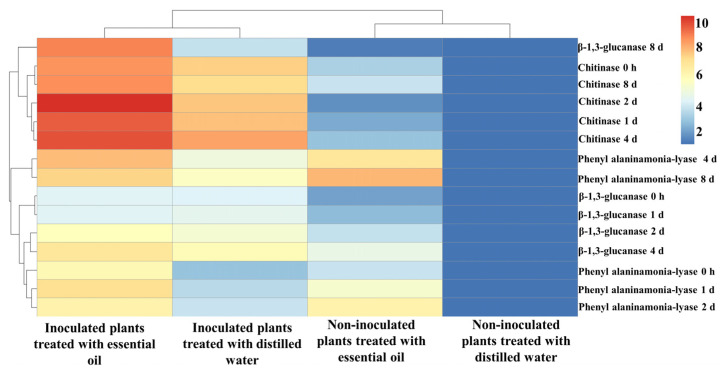
Heatmap of β-1,3-glucanase, chitinase, and phenylalanine ammonia-lyase gene expressions as a result of the treatment with celery essential oil (400 μg mL^−1^) in cucumber leaves inoculated with cucumber powdery mildew at different sampling intervals after the treatment with essential oil. The red color represents the maximum relative expression (fold change) of each gene, while the blue color represents the minimum relative gene expression compared to untreated healthy cucumber seedlings.

**Table 1 jof-10-00017-t001:** Chemical composition of essential oil from celery (*Apium graveolens*) seeds.

Compound	Molecular Formula	CAS	RI ^a^	RI ^b^	Rt ^c^ (Min)	TIC ^d^ (%)	Identification Method	References
β-Pinen	C_10_H_16_	000127-91-3	989	981	4.27	1.72	EI Mass, RI	[37]
β-Myrcene	C_10_H_16_	000123-35-3	996	991	4.58	1.07	EI Mass, RI	[38]
d-Limonene	C_10_H_16_	000138-86-3	1037	1035	5.77	42.74	EI Mass, RI	[39]
Pentylbenzene	C_11_H_16_	000538-68-1	1157	1158	10.03	2.38	EI Mass, RI	[40]
Pentanophenone	C_11_H_14_O	001009-14-9	1350	1359	18.44	1.48	EI Mass, RI	[41]
β-Selinene	C_15_H_24_	017066-67-0	1476	1479	23.86	11.91	EI Mass, RI	[42]
α-Selinene	C_15_H_24_	000473-13-2	1481	1485	24.1	1.7	EI Mass, RI	[43]
Caryophyllene oxide	C_15_H_24_O	001139-30-6	1568	1566	27.57	1.46	EI Mass, RI	[44]
β-Selinenol	C_15_H_26_O	000473-15-4	1638	1638	30.28	1.74	EI Mass, RI	[45]
3-Butylphthalide	C_12_H_14_O_2_	6066-49-5	1658	1658	31.08	14.42	EI Mass, RI	[46]
Mandelic acid	C_10_H_12_O_3_	000774-40-3	1731	- ^e^	33.72	10.92	EI Mass	^f^
Aromatics						29.20		
Monoterpenoids: Non-oxygenated						45.53		
Monoterpenoids: Oxygenated						- ^g^		
HSesquiterpenoids						16.81		
Minor compounds less than 1%						8.46		

^a^ Experimental retention index; ^b^ Reference retention index; ^c^ Retention time; ^d^ GC-MS total ion current; ^e^ Not found in NIST webbook library; ^f^ EI mass was compared with the standard in the NIST chemistry webbook library; ^g^ Not found.

## Data Availability

The datasets generated during and analyzed during the current study are in Appendix A or available from the corresponding author on reasonable request.

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
