# Peer review of "Defense Mechanisms Induced by Celery Seed Essential Oil against Powdery Mildew Incited by Podosphaera fusca in Cucumber"

_jof, 2023, doi:10.3390/jof10010017_

Round 1
Reviewer 1 Report
Comments and Suggestions for Authors
1. Line 63. Please add the full name of “PR1”.
2. Line 105. Please change “28 ºC” to “28 ℃”.
3. Line 114. Please add haemacytometer type, model and manufacturer used.
4. Line114. For sentence “Seedlings were inoculated by spraying the conidial suspension on the leaves”. Please specify the frequency of the spray applications.
5. Line118.The full name “CSEO” is not necessary at here.
6. Line 167,168,194. “Khan & Umar [30], Kar & Mishra [33], Livak and Schmittgen [36]”, the format should be unified.
7. Line187, 205. Please modify the paragraph format.
8. Line 207. Please change “CSO-treated” to “CSEO-treated”.
9. Line 216. Please delete “,” in “[42],”.
10. Line 271, 274, 301. Please change “phenols” to phenolic”.
11. Line 281. Please change “at 0 h, 1, and 2 d” to “at 0, 1, and 2 d”.
12. Figure 2. It is better to label the graphs with ABC for easy reading. The same below.
13. Line 318. For sentence “The activity of this enzyme was significantly higher in CSEO-treated than in untreated seedlings (Figure 3)”, the described results do not fully match those in Figure 3.
14. Line 340. It is better to delete “(0, 1, 2, 4, and 8 d after the treatment with essential oil)”.
15. Line 344. For sentence” In the two treated seedling sets, the activity of phenylalanine ammonia-lyase was higher in seedlings infected by powdery mildew at all sampling times (Figure 3)”, “two treated seedlings” should be replaced with “two untreated seedlings”. Please check it.
16. Line 374. Please change “0 h, and 2 d after the treatment” to “0 and 2 d after the treatment”.
17. Line 448. It is best to place Figure S2 in the main text, not in the supplements.
18. Line 475, 214, Table 1. The writing format of “3-butylphthalide” should be unified.
19. Line 516. Please add “,”, before “it can be hypothesized the increase of ……”.
20. Why it is difference for the flavonoid compounds contents in non-inoculated plants at 0 d.
21. Line 527. For sentence “Conversely, the level of flavonoid compounds did not show any change in untreated seedlings infected by powdery mildew compared to healthy seedlings”, the described results do not fully match those in Figure 2, such as the flavonoid compounds contents is significantly lower in untreated seedlings infected by powdery mildew than healthy plants at 1 d. Please check it.
22. Suggest to summarize the possible mechanism on how CSEO function in against powdery mildew in the last paragraph.
23. Line 627.check it please.
Author Response
Dear Reviewer,
our team appreciates the senior editor and the reviewers for their precious time and constructive review of our manuscript; we addressed all comments carefully. The corresponding changes and refinements made in the revised paper are summarized in our response below. The changes have been tracked using the ‘Track Changes’ feature and are highlighted in the text in the revised manuscript.
Yours sincerely

Reviewer 2 Report
Comments and Suggestions for Authors
Fig1. The figure shows data on the intensity of symptoms during treatment after inoculation. The only thing that is said about the effect of treatment before inoculation is that the before and after variants do not differ in Two-way analysis of variance. However, this difference does not mean a complete absence of nuance. It seems to me that these data should also be shown in the figure or, at a minimum, a joint text description of both options for the time of application of CSEO should be given, indicating that Fig1 shows only the data for the “after inoculation” option. In addition, it is not clear where the letter codings came from: did the authors conduct a one-way analysis of variance separately for each time variant of CSEO application?
In legends Fig2-4, in my opinion, it would be advisable to duplicate the information from the Methodology that the treatment was carried out 10 days after inoculation soon after the onset of symptoms, and the zero point is within one hour after the treatment.
Author Response
Dear reviewer,
our team appreciates the senior editor and the reviewers for their precious time and constructive review of our manuscript; we addressed all comments carefully. The corresponding changes and refinements made in the revised paper are summarized in our response below. The changes have been tracked using the ‘Track Changes’ feature and are highlighted in the text in the revised manuscript.
Yours sincerely
